# Endoscopic Ultrasound Advanced Techniques for Diagnosis of Gastrointestinal Stromal Tumours

**DOI:** 10.3390/cancers15041285

**Published:** 2023-02-17

**Authors:** Socrate Pallio, Stefano Francesco Crinò, Marcello Maida, Emanuele Sinagra, Vincenzo Francesco Tripodi, Antonio Facciorusso, Andrew Ofosu, Maria Cristina Conti Bellocchi, Endrit Shahini, Giuseppinella Melita

**Affiliations:** 1Department of Clinical and Experimental Medicine, University of Messina, 98100 Messina, Italy; 2Digestive Endoscopy Unit, University of Verona, 37100 Verona, Italy; 3Gastroenterology and Endoscopy Unit, S. Elia-Raimondi Hospital, 93100 Caltanissetta, Italy; 4Gastroenterology and Endoscopy Unit, Fondazione Istituto San Raffaele Giglio, 90015 Cefalù, Italy; 5Human Pathology of Adult and Child Department, University of Messina, 98100 Messina, Italy; 6Gastroenterology Unit, Department of Medical and Surgical Sciences, University of Foggia, 71100 Foggia, Italy; 7Division of Digestive Diseases, University of Cincinnati, Cincinnati, OH 45201, USA; 8Gastroenterology Unit, National Institute of Gastroenterology—IRCCS “Saverio de Bellis” Castellana Grotte, 70013 Castellana Grotte, Italy

**Keywords:** fine-needle aspiration, fine-needle biopsy, subepithelial lesions, artificial intelligence

## Abstract

**Simple Summary:**

Endoscopic ultrasound is now regarded as a valuable technique for assessing subepithelial lesions and determining their potential malignancy. The objective of this article is to thoroughly review the most recent advancements in the endoscopic field that can provide an accurate diagnosis and, as a result, establish the best treatment and outcomes for these types of lesions.

**Abstract:**

Gastrointestinal Stromal Tumors (GISTs) are subepithelial lesions (SELs) that commonly develop in the gastrointestinal tract. GISTs, unlike other SELs, can exhibit malignant behavior, so differential diagnosis is critical to the decision-making process. Endoscopic ultrasound (EUS) is considered the most accurate imaging method for diagnosing and differentiating SELs in the gastrointestinal tract by assessing the lesions precisely and evaluating their malignant risk. Due to their overlapping imaging characteristics, endosonographers may have difficulty distinguishing GISTs from other SELs using conventional EUS alone, and the collection of tissue samples from these lesions may be technically challenging. Even though it appears to be less effective in the case of smaller lesions, histology is now the gold standard for achieving a final diagnosis and avoiding unnecessary and invasive treatment for benign SELs. The use of enhanced EUS modalities and elastography has improved the diagnostic ability of EUS. Furthermore, recent advancements in artificial intelligence systems that use EUS images have allowed them to distinguish GISTs from other SELs, thereby improving their diagnostic accuracy.

## 1. Introduction

Gastrointestinal stromal tumours (GISTs) are the most common type of mesenchymal neoplasia that arises from the digestive tract [1]. Their histogenesis has been attributed to Cajal interstitial cells, which are thought to be the pacemaker cells of the gastrointestinal tract and are immunohistochemically positive for CD117 [2,3,4,5,6].

GISTs are more common in middle-aged (6th decade) males, with a prevalence of 14–20 cases per million, and are typically located in the gastric body (55.6%) or small intestine (31.8%) [1,2,3,4,5,6,7,8,9,10,11,12,13,14,15]. In 6.0% and less than 1% of cases, the colorectum and oesophagus are involved, respectively [7,8,9,10,11,12,13,14].

GIST-related complications are characterized by gastrointestinal bleeding (including acute melena and hematemesis, as well as chronic bleeding with subsequent anaemia) caused by mass ulceration, abdominal pain, weakness, and organ compression symptoms [3,4,5,12]. However, up to 30% of GISTs are incidentally discovered in asymptomatic patients or during routine examinations. They are typically uncovered as small subepithelial lesions (SELs) that are not ulcerated, are slightly elevated, and are covered by normal mucosa. Their subepithelial origin and commonly small size hamper their differentiation from other SELs, which have slow growth and an indolent course [10,15,16,17,18,19].

The diagnosis of a GIST relies on typical cell morphology (spindle cells) and immunohistochemistry, with strong reactivity for receptor tyrosine kinase KIT or CD34. Additional tests include DOG1 staining or mutation search of the KIT or PDGFRA genes [16]. 

GISTs have a known malignant potential, ranging between 10% and 30% [7,8,9,10,11]. The assessment of malignant potential allows for patient stratification according to very low, low, intermediate, or high-risk cases, which is necessary for the selection of treatment strategies [11]. Although the prognosis for patients with GISTs is mainly associated with the tumour size (>2 cm) and mitotic index (< or >5/50 HPF) [12,13], small GISTs with a low mitotic index can also have a malignant course with metastasis. Other prognostic factors include the primary tumour location, tumour rupture, and metastasis. 

When lesions are larger than 20 or 30 mm in diameter, surgical resection is the mainstay of treatment of localized GISTs [17]. Smaller tumours can be safely considered for endoscopic resection, with or without a laparoscopic control. However, despite complete resection, postoperative recurrence can occur in at least half of patients. Therefore, an early diagnosis is desirable [18,19,20,21,22]. 

Endoscopic Ultrasonography (EUS) is a crucial diagnostic technique for determining the potential malignancy of SELs, even though it is difficult to distinguish GISTs from other SELs using only EUS images [23,24]. The use of contrast agents or elastography in conjunction with EUS improves the latter’s diagnostic ability [25,26,27]. Furthermore, advances in artificial intelligence (AI) appear to have the potential to improve the accuracy of EUS for GIST diagnosis. However, although frequently controversial due to its associated technical difficulties and moderate diagnostic sensitivity, endoscopic biopsies or EUS-guided tissue acquisition (EUS-TA, including fine-needle aspiration (EUS-FNA) and fine-needle biopsy (EUS-FNB)) [28,29,30] continue to constitute the gold standard for making a definitive diagnosis. 

## 2. Endoscopic and EUS-Based Findings

The majority of SELs are asymptomatic and detected incidentally during endoscopy performed for unrelated causes. In general, their endoscopic appearance is typically characterized by a rounded protuberance with normal overlying mucosa, negative cushion signs, and, occasionally, a central depression or umbilication [31]. When GISTs increase in size, ulceration may become apparent. Spontaneous bleeding or fibrin clotting is associated with an increased risk of malignant transformation. 

Even when magnifying endoscopy or chromoendoscopy are used, SELs are extremely difficult to distinguish using solely conventional endoscopy. In general, attempts to differentiate GISTs from other SELs based on endoscopic findings have been inadequate with respect to small lesions. 

According to the European Society of Gastrointestinal Endoscopy (ESGE), EUS’s ability to define the morphology and features of the suspicion of malignancy render this technique the best diagnostic tool with which to characterize these lesions. EUS images show the location, size, originating layer (the fourth layer, which corresponds to the muscolaris propria), shape, internal echo pattern, heterogeneity, and vascularity of the lesion, as well as the presence of lymph nodes adjacent to or surrounding the tumour [31,32]. Several EUS features, including irregular borders, cystic spaces, ulceration, and echogenic foci, have been linked to a higher risk of malignancy (Figure 1). 

Furthermore, EUS-guided techniques such as contrast enhancement, elastography, and tissue acquisition have been investigated in terms of their ability to predict diagnosis and malignant behaviour. The key issue is distinguishing GISTs from other SELs. It is especially important to guide efficient clinical therapy regarding leiomyomas because GISTs are potentially malignant, whereas leiomyomas are benign [33]. Furthermore, when GISTs arise in the second/third portion of the duodenum with an extra-luminal extension, a differential diagnosis becomes more difficult, especially with respect to neuroendocrine tumours (NETs) (Table 1). Duodenal GISTs and NETs may look similar in imaging studies, and GISTs arising from the second or third portion of the duodenum may be misdiagnosed as pancreatic NETs based solely on imaging criteria. In addition, the resection techniques differ between these two tumours. Surgical excision with regional lymph node dissection is the best treatment for pancreatic NETs. GISTs, on the other hand, are frequently treated with minimal resection and without lymph node dissection. Hence, the role of histological diagnosis is critical in determining their appropriate treatment and outcomes [31,32,33,34,35].

As SELs are located in the inner layer, with overlying normal mucosa and submucosa, the diagnostic yield of conventional endoscopic forceps-based biopsy is limited, ranging from 17% to 59%, despite the use of special devices such as the “jumbo” forceps or dedicated techniques such as the “bite-on-bite” biopsies. To address this limitation, the mucosal incision-assisted biopsy (MIAB) was developed. This technique entails lifting the mucosa that covers the SEL to make a more secure incision. The exposed lesion is sampled with biopsy forceps after an electrosurgical incision of the target mucosal and submucosal tissues with an endoscopic submucosal dissection knife [36,37,38,39,40,41].

As previously stated, EUS-based tissue acquisition, either FNA or FNB, is a viable alternative with a diagnostic rate ranging between 71% and 100%, which is strongly influenced by tumour size. The ability to perform a mitotic count, the risk of seeding, and the feasibility of the technique in specific sites were all observed to be critical issues.

Although EUS-FNB and MIAB are both recommended by ESGE guidelines, the procedure time, the size and location of the SELs, and expertise influence the choice of procedure. Tissue diagnosis is recommended for all SELs with GIST-like characteristics that are larger than 20 mm, have high-risk stigmata, or require surgical resection or oncological treatment.

## 3. Contrast-Enhanced Harmonic EUS

The use of contrast agents has improved the diagnostic performance of EUS, particularly with respect to differentiating GISTs from other gastrointestinal SELs [42].

Conventional EUS B-mode analysis is performed to assess the size and shape of SELs, their origin wall layer, and ultrasonographic characteristics (tissue echogenicity, calcifications, vascularization, or the presence of avascular areas using Power Doppler or hi-flow). Contrast-enhanced harmonic EUS (CH-EUS) can then visualize the microvascularization of SELs, enhancing their characterization, with hyperenhancement specific to GIST and hypo-enhancement specific to benign SELs. 

When exposed to an ultrasonic wave, the contrast agents oscillate or break [42]. SonoVue (Bracco SpA, Milan, Italy) and Sonozaid (Daiichin-Sankyo, Tokyo, Japan) are contrast media that contain safe microbubbles covered by a protective lipophilic shell that carries carbon dioxide gas. In response to acoustic stimuli, these bubbles oscillate, thereby increasing the echo levels in the target tissue. During CH-EUS, the optimal amount of contrast medium is injected intravenously while the ultrasound machine is in contrast-harmonic mode. When performing CH-EUS with the SonoVue^®^ contrast agent, a 4.8 mL bolus of SonoVue^®^ is injected through a peripheral intravenous cannula, followed by a 10 mL saline flush [43]. Each patient’s contrast study usually lasts 90 s after the intravenous bolus injection and is documented by a video clip that includes B-mode examination and the arterial, portal, and late phases.

Contrast enhancement is typically evaluated in the early (after a few seconds) and late phases (after more than 30 s), and the enhancement patterns are then classified (as hyper-, iso-, or hypo-enhancement, and as homogeneous or inhomogeneous) along with the features (the presence or absence of regular or irregular intratumoral vessels, and the presence or absence of an unenhanced area) that can be observed after the injection of the contrast medium [44].

Pancreatic diseases are the primary application field for CH-EUS [45]. The pooled sensitivity and specificity of CE-EUS with respect to distinguishing pancreatic cancers from solid inflammatory masses were reported as 93% and 88%, respectively, in a 2017 meta-analysis [45]. Moreover, CH-EUS is also recommended for investigating pancreatic cysts, gallbladder and biliary tract lesions, lymph nodes, and SELs [46,47,48].

Several studies [31,44,49,50,51,52,53,54,55] have found CH-EUS to be useful for the characterization of GISTs. The pooled sensitivity and specificity were 89% (95%CI 82–93%) and 82% (95%CI 66–92%), respectively, in a meta-analysis [44] published in 2019 that included seven studies [49,50,51,52,53,54,55] with a total of 187 patients and assessed the value of CH-EUS towards distinguishing between GISTs and other benign SELs. One limitation of this meta-analysis was the inclusion of only two prospective studies [49,50]. The first [50], an international multicentre study, compared GISTs with leiomyoma using the CH-EUS-based characterization of 62 SELs in different locations in the upper gastrointestinal tract. Despite the small number of benign SELs discovered (5 leiomyomas vs. 57 GISTs), CH-EUS revealed hyperenhancement and avascular areas in a high percentage of GISTs but not in leiomyomas. However, the lesion size was not uniform (mean size 62.6 ± 42.1 with a range from 16 to 200), and there was a trend toward a smaller size for GISTs without avascular areas (65.8 ± 43 (16–200) vs. 39.6 ± 26.9 (22–90) *p* = 0.062). Moreover, there was no attempt to stratify malignant potential. In the second study, Sakamoto et al. used microvasculature evaluation with intratumoral vessel quantification (regular pattern, irregular pattern, or absence of vessels) to characterize 29 GISTs, and compared the results to histological or surgical specimen diagnosis and malignancy assessment. Similarly, many studies reported sensitivity and specificity ranging from 75% to 100% and 63% to 100%, respectively [44,49,52]. Sakamoto et al. demonstrated that an irregular intratumoral vessel pattern was an 83% accurate predictor of high-grade malignant GISTs [49] (Figure 2).

According to Tamura T and Kitano M’s 2019 review, CH-EUS can distinguish between GISTs and other gastrointestinal SELs with sensitivity and specificity ranging from 78–100% and 60–100%, respectively [43].

In addition, in another study not included in the mentioned meta-analysis [49], CH-EUS was used on 14 patients with SELs of the stomach (11) and oesophagus (3), of which most originated in the fourth layer. Leiomyoma was the final diagnosis in four cases, GISTs in five, schwannoma in one, and other rare lesions in four. All GISTs were hyperenhanced, while all other tumours except one leiomyoma were hypo-enhanced [46]. The main findings via Ch-EUS are summarized in Table 2. 

Even though CH-EUS improves the accuracy of EUS towards SEL characterization, it cannot replace tissue acquisition for differentiating GISTs from other spindle cell neoplasms (leiomyomas), which share a “regular” vessel pattern. As a result, histology must be used to assess the malignant potential of GISTs.

## 4. EUS-Elastography

EUS-E is a real-time imaging technique that analyses tissue elasticity and displays this information graphically as a colour spectrum of shades [59]. While green represents average stiffness, blue represents harder tissue and red represents softer tissue. Each colour is associated with a specific value of tissue elasticity in a defined region of interest, ranging from 1 to 255 kPa [59] (Figure 3).

In addition, EUS-E compares the strain between the target and other reference areas, delivering a semi-quantitative analysis of tissue stiffness [60]. In more detail, strain ratio (SR) is a value derived from the ratio of the stiffness of two user-defined areas within an elastogram that provides an objective estimation of the lesion’s hardness [61].

Due to its lack of invasiveness, EUS-E was initially used for the differential diagnosis of SELs by providing a qualitative/semi-quantitative stiffness analysis [58,62,63,64,65,66,67,68,69,70]. EUS-E is an imaging technique that detects diseased and normal tissue elasticity changes on conventional B-mode ultrasound images [70]. The fundamental principle of EUS-E imaging is that tissues have varying elasticity, thus causing different strains when compressed by an external force or when compressed by normal breathing and blood circulation. An ultrasonic system’s software program can then characterize and visualize these strain values in real-time [71]. The elasticity values are then visually characterized in different colours on the elastography images based on tissue deformation [50,64]. Elastography is commonly used in clinical practice to diagnose diseases of the liver, thyroid, kidney, lymph nodes, prostate, mammary glands, and pancreas [34,69,72]. 

However, to date, only a few studies have examined the role of EUS-E in the diagnosis of SELs in the gastrointestinal tract, and the results are still debatable. The first pilot study on the efficacy of EUS-E for differentiating 25 consecutive gastric SELs demonstrated that GISTs were qualitatively harder than other SELs by rating the degree of stiffness based on the majority and colour distribution [62]. As proven by a prior study, GISTs tend to have a blue colour (61/62, 98%), which was confirmed by a subsequent study. This tendency, however, resembles that of leiomyoma (4/5, 80%) [50]. The feasibility of quantitative EUS-E based on SR with respect to the differential diagnosis of SELs has been investigated since its introduction. A preliminary retrospective study of 30 patients found that EUS-E with SR may be promising in terms of differentiating GI SELs, wherein a cut-off of 11.18 can distinguish between GISTs and leiomyomas with a sensitivity of 81.8% and a specificity of 85.7% [73]. In a prospective study by Kim et al., the SR of 41 gastric SELs was compared with the histopathologic diagnosis. GISTs presented an elevated SR (mean 51.1 ± 11.1) that enabled them to be distinguished from leiomyoma, whose SR was considerably lower (6.0 ± 6.9), with a favourable sensitivity of 100% and a specificity of 94.1% when using an SR cut-off of 22.7 [63]. The distinction of GISTs from schwannoma, a mesenchymal tumour with a similar appearance to a GIST, appears to be challenging, given that the mean SR of the only schwannoma in the series was 62.0 [32]. In a recent study, Guo et al. utilized hue histograms to quantify EUS-E images but did not find adequate evidence to support the utility of EUS-E for differentiating between GISTs and gastrointestinal leiomyomas using EUS-E [66]. 

As a result, EUS-E is seen as a promising less-invasive diagnostic modality; however, the complete discrimination of GISTs from other SELs in a single use of elastography remains challenging, and more robust data are required to assess the efficacy of EUS-E (Table 3). Ultimately, EUS-E can estimate valuable information and enhance diagnostic accuracy for patients with gastric SELs. However, there is a boundary. The size and number of mitoses per 50 high-power fields are used to assess GIST malignancy. Elastography can indicate the existence of a GIST but not its malignant possibility [74].

## 5. EUS-Guided Fine-Needle Tissue Acquisition

EUS-FNA is the most established method of SEL tissue sampling and can provide a conclusive cytological and immunohistochemical diagnosis safely and reliably. The accuracy of EUS-FNA is highly dependent on the availability of rapid on-site evaluation (ROSE) as well as other criteria such as endoscopists’ and pathologists’ experience, the target lesion’s characteristics, and tissue handling and processing. To compensate for the limited availability of ROSE in many locations, recent advances in EUS-FNB needle technology have significantly increased the capacity to collect histology samples in up to 90% of patients [72,75,76,77,78]. 

Lesions can be punctured with a histology needle under the guidance of EUS, with the option of switching to a needle of a different size if technically or clinically indicated. Based on prior investigations, which showed that gastric SELs have an 83% sample adequacy with 2.5 needle passes and a diagnostic accuracy plateau with 2.5–4 needle passes [79,80,81], the minimum number of needle passes is set at 3 to obtain a sufficient amount of material for both pathological and immunohistochemistry analysis. When an on-site cytological evaluation is not possible, the 2017 ESGE guidelines recommended the use of three to four needle passes with an FNA needle or two to three passes with an FNB needle [82]. 

Several randomized controlled trials and retrospective studies [83,84,85] have demonstrated the superiority of EUS-FNB over EUS-FNA, which has led to the abandonment of ROSE [86].

Notably, a recent study recommended at least three passes with 22-gauge ProCore needles during EUS-FNB using the standard suction technique and at least four passes using the slow-pull technique. In terms of tissue acquisition and diagnostic capabilities, the standard technique demonstrated potential advantages over the slow-pull technique [87]. With regard to 57 patients, a 2022 Japanese study [88] compared the usefulness of 22G Fork-tip and Franseen needles for EUS-TA and assessed the ability of CH-EUS to diagnose SELs ≤ 2 cm. The rate of adequate sample acquisition with Fork-tip needles was significantly higher than with Franseen needles (96% vs. 74%; *p* = 0.038). The presence of a hyper- or iso-vascular pattern upon conducting CH-EUS was significantly correlated with the presence of a gastrointestinal stromal tumor (*p* < 0.001). In terms of sample acquisition, EUS-TA with Fork-tip needles outperformed the use of EUS-TA with Franseen needles, and CH-EUS was also useful for the diagnosis of SELs ≤ 2 cm.

The management of SELs and GISTs remains difficult. For 3 cm gastric SELs without considering endosonographic features, the American Gastroenterology Association Institute technical review recommended a follow-up using EUS or endoscopy at regular intervals [89]. Due to their malignant potential, GISTs > 2 cm should be surgically resected according to the National Comprehensive Cancer Network. Endosonography can be used to monitor small gastric SELs with no high-risk features, according to recent guidelines [90,91,92,93]. 

Although endosonographic imaging features have been assessed as being able to predict SELs’ malignant potential, EUS-TA has been accepted as a standard technique for obtaining tissue, regardless of whether forward- or oblique-viewing echoendoscopes are used. Indeed, a prospective, randomized trial with a cross-over design enrolled 41 patients with SELs who underwent EUS-FNA with two echoendoscopes. The histological yield was similar when using forward-viewing and oblique-viewing scopes (80.5% vs. 73.2%, *p* = 0.453). Moreover, similar tissue area and rates of sampling success were observed [94]. 

Moreover, a study including eight patients evaluated the feasibility and safety of performing EUS-FNA using a forward-viewing echoendoscope in patients with small (approximately 10 mm) SELs [80]. The authors found that, with the use of a cap, even the smallest lesions were easily targeted, obtaining seven out of eight (87.5%) adequate samples with no adverse events. The development of drill needles is another innovation in the field of EUS-TA that is expected to improve the diagnostic rates for the detection of even small SELs [95]. Uesato et al. evaluated 13 consecutive resected gastric specimens containing SELs (11 GISTs, 1 Schwannoma, and 1 ectopic pancreas) [96]. All lesions were sampled using a drill and standard FNA needles. Using the drill needles, 100% histological specimens were obtained, which resulted significantly better when compared with the value of 61.5% obtained using FNA needles (*p* = 0.047) [96]. Similar results were obtained with respect to small (<25 mm) lesions (7/7, 100% vs. 3/7, 42.9%, *p* = 0.035) [96]. However, such needles’ effectiveness in clinical practice and under EUS guidance remains to be demonstrated.

The typical EUS-FNA findings with respect to GISTs are spindle-shaped cells or epithelial cells that are positive for KIT or CD34. For SELs, the EUS-FNA diagnosis rate ranges from 62.0% to 93.4% [79,96,97,98]. The diagnosis rate for 1 to 2 cm tumours is 71%, while this rate is 86% for 2 to 4 cm tumours, and 100% for tumours larger than 4 cm [97]. As the tumour’s diameter increases, so does the rate of diagnosis. Unfortunately, using a standard EUS-FNA scope on a subepithelial hypoechoic solid mass of 1 cm is technically challenging; thus, EUS-FNA is recommended for masses greater than 1 cm [96,99]. Furthermore, mitosis evaluation is important for determining the metastatic risk of GISTs because a high Ki-67 labelling index is associated with a higher risk of recurrence and poor survival [99]. Unfortunately, the volume of a tissue sample obtained by EUS-FNA is usually low. As a result, assessing mitosis using EUS-FNA is difficult and not recommended by guidelines. However, different results were reported by some authors. For example, Ando et al. [100] reported that the MIB-1 labelling index is 100% accurate for the diagnosis of malignant GISTs because Ki67-positive cells can be easily identified in small EUS-FNA specimens. 

In a meta-analysis of 10 studies with 669 patients [101] comparing EUS-FNA and EUS-FNB, EUS-FNB outperformed EUS-FNA for all the diagnostic outcomes evaluated, including in terms of adequate sample rate, optimal histologic core procurement rate, diagnostic acuity, and the number of passes required to obtain diagnostic samples. The majority of the needles utilized were 22G, and the examined EUS-FNB needle designs included reverse-bevel ProCore™ (Cook Medical, Limerick, Ireland), Acquire™ (Boston Scientific, Scientific, Marlborough, MA, USA), and SharkCore™ (Medtronic, Boston, MA, USA). The observed adverse events, mostly minor bleeding, were rare (<1%) and occurred when EUS-FNB and EUS-FNA were used, thus demonstrating that EUS-FNA/B is a first-line technique for obtaining tissue samples to determine the histology of SELs, although it is less effective for small lesions [6,31,93]. 

Furthermore, as with other abdominal malignancies, some authors have speculated about the risk of seeding. However, a recent meta-analysis [101] revealed that the relationship between biopsy and GIST recurrence was never evaluated in a prospective trial, and that seeding is generally linked to tumour biology and the technique of the biopsy itself (mainly for the percutaneous approach). As a result, EUS-TA can be used safely in these tumours to confirm the diagnosis or distinguish these lesions from other gastric cancers, thus guiding treatment. With regard to all the reasons stated above, the recommendations from various guidelines differ. Although it is sometimes challenging, the European Society for Medical Oncology [91] and the Japanese GIST guideline Subcommittee [92] recommend surgical resection for all SELs immunohistologically diagnosed as a GIST. Accordingly, the ESGE guidelines for the management of SELs [31] suggest providing tissue from all SELs with features suggestive of GIST or for SELs larger than 20 mm. Given the superiority of EUS-FNB over EUS-FNA, the ESGE [31] recommends using either EUS-FNB or mucosal incision-assisted biopsy if SELs are larger than 20 mm. In the case of smaller lesions (less than 20 mm), mucosal incision-assisted biopsy should be used as the first-line technique.

## 6. Artificial Intelligence

Distinguishing GISTs from other non-GISTs based on EUS imaging without histological data is sometimes challenging due to the observers’ subjective interpretations of the EUS images and limited interobserver agreement (Table 4).

Even though endoscopy and EUS provide information on the layer of origin and location of a lesion, such as the cardia or body of the stomach, it is difficult to identify GISTs with EUS (B-mode) alone, with the sensitivity and specificity for human physicians being 75.8% and 85.5%, respectively [108,109]. 

In recent years, Artificial intelligence (AI) based on deep learning techniques such as convolutional neural network (CNN) has advanced dramatically in the medical field and is now used to improve diagnostic accuracy in gastrointestinal endoscopy. It may be useful for risk stratification and facilitating clinical management, in which it leads to improved health outcomes [80,110].

The development of EUS-AI entails the incorporation of EUS images of histologically confirmed GISTs and SELs into an internal dataset. AI systems can be trained to recognize “normal” characteristics by associating a gold standard with appropriate images using machine learning and, more recently, deep learning [111]. Early AI evolution is a result of the development of an algorithm that employs extremely accurate datasets created and arranged independently by a team of specialists [112,113]. Although the results of previous studies remain controversial, AI-based diagnostic systems designed for EUS have shown good performances.

EUS-AI diagnostics can be performed automatically and safely and provide rapid diagnoses, gathering pixel-level information that are invisible to the naked eye [79,113,114,115] without using invasive sampling techniques such as EUS-FNA or EUS-FNB. EUS-AI diagnoses can be made objectively, whereas EUS expert diagnoses are likely to be interpreted subjectively. Non-experts may be able to diagnose GISTs differentially with the same or greater accuracy than EUS experts with the help of EUS-AI [23,116].

Following published standards, stacking or “bite-on-bite” pinch biopsies may be used for SELs; however, their accuracy is frequently low [117,118]. In addition, even with EUS-FNB, obtaining acceptable specimens for SELS above 20 mm is more difficult and may result in low diagnostic sensitivity [105,107,119,120]. Consequently, the ESGE guidelines for EUS-guided samples do not propose EUS-FNA for patients with SELs less than 20 mm [97]. In contrast, the AI system may significantly increase the diagnostic accuracy of SELs without the restriction of SEL size and the need for invasive biopsy [121]. 

In addition, the combination of endosonographers’ diagnosis and AI similarly improved the diagnostic accuracy of SELs utilizing EUS. 

Yang et al. investigated the diagnostic capabilities of AI with respect to EUS images and discovered that the use of AI diagnosis increased the accuracy of distinguishing GISTs from leiomyoma from 73.8% to 88.2% when compared to expert endosonographers’ diagnosis. According to the authors, AI systems are expected to reduce the rate of GIST and SEL misdiagnosis, thereby assisting patients in avoiding unnecessary EUS, invasive biopsies, and surgeries [105].

Thirty sets of EUS images with SELs higher or lower than 20 mm were prepared for diagnosis by an EUS diagnostic system with AI and three EUS experts in a 2020 Japanese pilot study [115]. In comparison, the accuracy, sensitivity, and specificity for SELs ≥ 20 mm for the EUS-AI were 90.0, 91.7, and 83.3%, respectively, and 53.3, 50.0, and 83.3%, respectively, for the EUS experts. The AUC for the diagnostic yield of the EUS-AI for SELs ≥ 20 mm was significantly higher than that of the EUS experts (0.965 vs. 0.684, respectively; *p* = 0.007).

Hirai K’s 2022 multicentre study [104] investigated the efficacy of an AI system for the clarification of SELs in EUS images. For the development and test datasets, a total of 16,110 images were collected from 631 cases. The AI system’s sensitivity, specificity, and accuracy for distinguishing GISTs from non-GISTs were 98.8%, 67.6%, and 89.3%, respectively. Its sensitivity and accuracy were significantly higher than those of all the endoscopists combined. As a result, the AI system classifying SELs outperformed the experts in terms of diagnostic performance.

Notably, Tanaka H et al. [106] conducted a retrospective study among 53 patients with GISTs and leiomyomas to assess the value of AI in the diagnosis of gastric SELs by CH-EUS. SiamMask, a novel technology, was used to track and trim lesions in CH-EUS videos. Deep learning with a residual neural network and leave-one-out cross-validation were used to evaluate CH-EUS. The sensitivity, specificity, and accuracy of AI with respect to diagnosing GIST were similar compared to blind reading (90.5%, 90.9%, and 90.6% vs. 90.5%, 81.8%, and 88.7%, respectively; *p* = 0.683). The coefficient of correlation between the two reviewers was 0.713. Finally, the diagnostic ability of the AI-evaluated CH-EUS results to distinguish between GISTs and leiomyomas was comparable to that of blind reading by expert endosonographers.

A 2022 systematic review and meta-analysis [107] aiming to assess the diagnostic accuracy of AI-based EUS for distinguishing GISTs from other SELs identified eight studies in relation to this task. With regard to differentiating GISTs from leiomyoma, the AI model showed a pooled AUC of 0.94, a sensitivity of 93%, and a specificity of 78%. Similar results were observed in the results obtained by applying only CNN models, whereas a slightly lower specificity (72%) was reached using only B-mode EUS images. 

Another study not included in this meta-analysis was recently published [121]. Zhu et al. developed an AI system able to detect protruding benign gastric lesions during standard white light endoscopy and implement EUS examination. They enrolled more than 1300 patients with SELs who underwent upper gastrointestinal endoscopy and/or EUS. Qualified images were screened by two expert endoscopists and included in the dataset. The two unimodality models were then merged into a single hybrid model including both white light endoscopy and EUS. The hybrid system achieved an area under the curve (AUC) of 0.89, 0.99, and 0.89 for leiomyomas, gastric ectopic pancreas, and GISTs, respectively, reaching an accuracy of up to 86.6% for distinguishing GISTs from other SELs. Moreover, the system’s performance was compared to that of twelve endoscopists with varying levels of expertise who were blinded to clinical information. The hybrid model outperformed endoscopists’ accuracy in terms of GISTs (83.5% vs. 71%) and leiomyoma (78.5% vs. 72%), whereas the accuracy for gastric ectopic pancreas was comparable (98.3% vs. 97.8%). Thus, the system was validated both internally and externally.

A dataset containing information from 1366 participants was used in another recent Chinese study [121] to train and validate a multimodal, multipath AI system for classifying GISTs, which achieved the highest AUC among the tested methods of 0.896. The performance of the model was validated using both external and internal longitudinal datasets. In terms of SEL recognition accuracy, the multipath AI system outperformed expert endoscopists.

Finally, aside from CNN models, computer evaluation of EUS images has been recently attempted, even using widespread image-processing software such as Photoshop [122]. The authors included 472 patients with 239 gastric leiomyomas and 233 gastric GISTs. They used Photoshop to calculate the mean grey value of the tumours, muscularis propria, submucosa, and water. Furthermore, the ratio of the mean grey value of the tumour to the muscularis propria, submucosa, and water was also calculated. The mean grey value of tumours was significantly higher in GISTs than in leiomyomas, with an AUC of 0.952 (95%CI 0.897–1.000), a sensitivity of 90%, and a specificity of 97.5%. Similarly, the tumour/muscolaris propria ratio, the tumour/submucosa ratio, and the tumour/water ratio were all significantly higher in the GIST group, and their AUC values were 0.917 (95%CI 0.844–0.991), 0.897 (95%CI 0.812–0.981), and 0.929 (95%CI 0.887–0.987), respectively. The sensitivity and specificity of the tumour/muscolaris propria ratio, the tumour/submucosa ratio, and the tumour/water ratio were 92.5% and 95%, 90% and 92.5%, and 87.5% and 92.5%, respectively.

While these AI systems are highly effective, they have some limitations. Only a few EUS probes are useful with an AI model. Analysing the pixel values of EUS photos reveals that imaging discrepancies between EUS probes can occur, which may impact AI performance. Likewise, these AI systems were designed to distinguish GISTs from SELs and cannot be effectively applied to other rare diseases such as leiomyosarcoma, Schwannomas, or glomus tumours. Further prospective clinical trials on the diagnosis of GISTs with AI-EUS are needed.

## 7. Conclusions

GISTs’ differential diagnosis can be difficult at times. EUS is required to collect information about the target and to assess the possibility of malignancy. 

The use of contrast media or elastography in conjunction with EUS appears to improve the latter’s diagnostic capability. The development and implementation of non-invasive methods, such as AI-assisted diagnosis, are expected to provide an alternative to invasive, histological diagnosis, which is currently the gold standard. 

Nonetheless, although contemporary advances in AI may aid the detection of GIST in clinical practice, more prospective clinical studies on the diagnosis of GISTs using AI-based EUS are required to confirm the earlier discoveries.

## Figures and Tables

**Figure 1 cancers-15-01285-f001:**
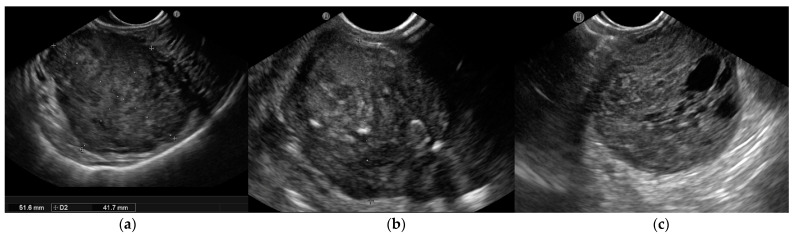
Endoscopic ultrasound (EUS) images of malignant gastrointestinal stromal tumors (GISTs): (**a**) A large submucosal lesion originating from the fourth layer of the gastric wall. Echopattern is inhomogeneous with irregular borders. (**b**) Another large subepithelial gastric mass with echoic foci, calcifications, and irregular profiles. (**c**) Cystic spaces are visible in EUS images.

**Figure 2 cancers-15-01285-f002:**
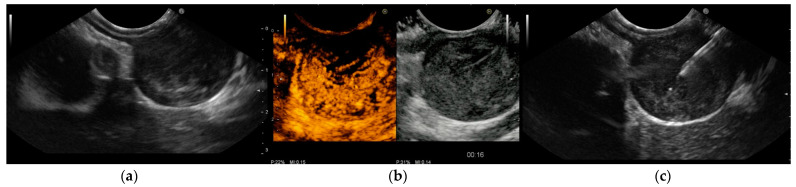
Endoscopic ultrasound (EUS) images of gastrointestinal stromal tumor (GIST) of the stomach: (**a**) The originating layer is visible when the ultrasound transducer is placed at the peripheral portion of the lesion. (**b**) Contrast-enhanced harmonic EUS (CH-EUS) demonstrated a hypervascular pattern. Moreover, CH-EUS allowed for the identification of irregular large vessels and avascular areas inside the tumor. (**c**) EUS-guided fine-needle biopsy was performed using a 22-gauge end-cutting needle while trying to avoid avascular areas previously defined using CH-EUS. Histology confirmed a GIST with a high replicative index.

**Figure 3 cancers-15-01285-f003:**
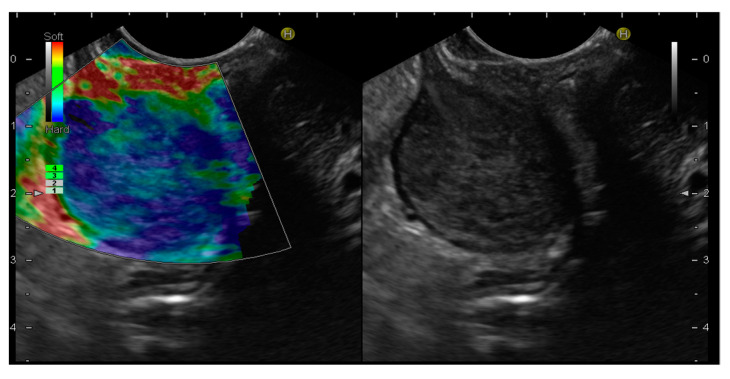
Endoscopic ultrasound elastography (EUS-E) images of gastrointestinal stromal tumor (GIST) of the stomach. The lesion shows a blue color, indicative of a hard tissue, compared to the red color of the gastric wall.

**Table 1 cancers-15-01285-t001:** EUS features of subepithelial lesions (SELs) in the gastrointestinal tract.

SEL Type	Originating Layer	Echogenicity	Size (mm)	Border	Location in Gastrointestinal Tract
Duplication cyst	3rd		-	Sharp	Any
Varices	3rd	Anechoic, with doppler signal	-	Sharp, serpiginous shape	Any
Gastric inflammatory polyp	2nd, 3rd	Hypoechoic, homogeneous, polypoid	8–20	Variable	Antrum Small bowel
Neuroendocrine tumor	2nd, 3rd	Hypoechoic, intermediate hypoechogenicity, Hyperechoic	Variable	Sharp	Stomach Small bowel Rectum
Ectopic pancreas	3rd, 4th	hypoechoic, heterogeneus echotexture, cyst or duct inside, central umbilication.	<5–20	Variable	Antrum Gastric body Duodenum
Leyomioma	2nd, 4th	Hypoechoic, homogeneus.	Variable	Sharp	Esophagus, Stomach, Anywhere in GI tract
GIST low risk	2nd/4th	Hypoechoic, homogeneus, hypervascular.	<30	Regular	Esophagus, Stomach, Small Intestine, Rectum
GIST high risk	2nd/4th	Hypoecoic, heterogeneus cystic space, echogenic foci, calcifications, dimpling or ulcers.	>30	Irregular	Esophagus, Stomach, Small Intestine, Rectum
Lymphoma	2nd, 3rd, 4th	Hypoechoic	Variable	Irregular	Gastric, Small intestine
Schwannoma	4th	Hypoechoic, homogeneous, marginal halo.	-	Sharp	Gastric body
Lipoma	3rd	Hyperechoic, Homogeneous.	-	Irregular	Any

**Table 2 cancers-15-01285-t002:** Studies on the ability of Contrast-Enhanced Harmonic Endoscopic Ultrasound (CEUS-E) to detect Gastrointestinal Stromal Tumors (GISTs).

Author	Study	N. GISTs	Lesion Size mm	Echo Pattern	Sensitivity	Specificity	PPV	NPV	AUROC	Conclusion
Sakamoto et al., 2011 [49]	Prospective	29 (*n* = 29 pts)	>30 mm (18/29, 62%)	Type I (regular vessels, homogeneous enhancement): Low-grade malignancy (*n* = 8); Type II (irregular vessels, heterogeneous enhancement): High-grade malignancy (*n* = 16), low-grade malignancy (*n* = 5)	100% (malignancy prediction based on irregular vessels)	63% (malignancy prediction based on irregular vessels)	NA	NA	83% (malignancy prediction based on irregular vessels)	CH-EUS successfully visualized intratumoral vessels and may be useful in predicting GIST malignancy risk
Yamashita Y et al., 2015 [52]	Prospective	13 (*n* = 13 pts)	1.9–60	Hyperenhancement (*n* = 13/13); vessel positive (*n* = 6): very low/ low-grade malignancy, 1 (17%); Intermediate/high-grade malignancy, 5 (83%)—vessel negative (*n* = 7): very low / low-grade malignancy, 7 (100%)	NA	The specificity of rich vascularity determined via CE-EUS for intermediate or high-risk GIST was high	NA	NA	NA	Intratumoral vessels identified using CE-EUS in GISTs are associated with a higher degree of angiogenesis, implying a higher malignant potential
Park HY et al., 2016 [55]	Retrospective	35	32.5 ± 12.5	Irregular vessels: high-grade malignancy (63.6%), low-grade malignancy (46.7%); Heterogeneous perfusion: high-grade malignancy (36.4%), low-grade malignancy (26.7%); Non-enhancing spots: high-grade malignancy (63.6%), low-grade malignancy (46.7%)	53.8%	66.7% [N. positive findings > 1 (benign vs. GIST)]	86.4% [N. positive findings > 1 (benign vs. GIST)]	46.2% [N. positive findings > 1 (benign vs. GIST)]	71.4% [N. positive findings > 1 (benign vs. GIST)]; 63.6% (malignancy prediction)	CH-EUS had low sensitivity, specificity, and accuracy in predicting SEL malignancy risk
Ignee A et al., 2017 [50]	Prospective	57 (SELs, *n* = 62)	62.6 ± 42.1 (16–200)	Hyperenhancement: 56/57 (98%); avascular areas: 50/57 patients (88%)	98%	100%	100%	93%	98%	CH-EUS reveals hyperenhancement and avascular areas in a high percentage of GISTs but not in leiomyoma. GISTs and leiomyoma can thus be distinguished precisely
Kannengiesser K et al., 2017 [51]	Prospective	8 (*n* = 17 pts)	NA	Hyperenhancement (maximum intensity, 47.3 ± 11.6 db) (*n* = 8/8)	NA	NA	NA	NA	NA	CH-EUS can accurately distinguish GISTs from benign lesions
Kamata K et al., 2017 [54]	Retrospective	58 (*n* = 73 pts)	28 (10–90)	Hyperenhancement: 49/58 (84.5%); inhomogeneous: 21/58 (36.2%)	84.5%	73.3%	NA	NA	82.2%	GISTs were discovered to have hyper-enhancement and inhomogeneous enhancement
Pesenti C et al., 2019 [46]	Retrospective	5 (SELs, *n* = 14)	35	Hyperenhancement: 5/5 (100%)	100%	NA	NA	NA	NA	CH-EUS could be used in conjunction with EUS to differentiate GISTs from other SELs (early and clear enhancement)
Cho IR et al., 2019 [56]	Retrospective	37 (*n* = 176 pts)	2.61 ± 1.71	Hyperenhancement: 51.4%; positive vascularity: 81.1%; lower LSR: 1.3	81.1% (vascularity)	84.8% (vascularity)	85.8% (vascularity)	80% (vascularity)	82.9% (vascularity)	Upon conducting CH-EUS, the LSR and vascularity of SELs can be used as parameters for a noninvasive GIST prediction model
Tang JY et al., 2019 [44]	Meta-analysis	*n* = 187 pts	25–63	Hyperenhancement: 100%	89% (95%CI 0.82–0.93)	82% (95%CI 0.66–0.92)	NA	NA	0.89	CH-EUS is a noninvasive, safe method for differentiating GIST from benign SELs and, to a lesser extent, predicting their malignant potential
Lee HS et al., 2019 [57]	Retrospective	32 (*n* = 44 pts)	Low-grade malignancy: 27 (16–50); High-grade malignancy: 34 (15–65)	Low-grade malignancy: irregular vessels 11 (55.0), heterogeneous perfusion 12 (60.0), hyperechoic foci 10 (50.0), non-enhancing spots 11 (55.0); High-grade malignancy: irregular vessels 8 (66.7), heterogeneous perfusion 5 (6.2), hyperechoic foci 8 (66.7), non-enhancing spots 8 (66.7)	84.4% (perfusion)	60% (perfusion)	93.1% (perfusion)	37.5% (perfusion)	NA	The combination of CH-EUS and perfusion analysis performed with perfusion analysis software may be a quantitative and independent method for predicting malignancy risk in gastrointestinal SELs
Lefort C et al., 2021 [58]	Retrospective	40 (*n* = 54 pts)	40 (15–150)	Hyperenhancement (NA)	Diagnostic (GIST): 85%; malignancy GISTs 100%	Diagnostic (GIST): 57.1%; malignancy prediction: 82.1%	NA	NA	Diagnostic (GIST): 77.8%; malignancy prediction: 86.1%	CH-EUS outperformed B-mode EUS with respect to differentiating leiomyomas and risk stratifying GIST. The addition of CH-EUS improved diagnostic accuracy in high-grade GISTs

Abbreviations: GISTs: Gastrointestinal stromal tumors; SELs: Subepithelial lesions; GI: gastrointestinal; LSR: long-to-short axis ratio.

**Table 3 cancers-15-01285-t003:** Studies on the use of Endoscopic Ultrasound Elastography (EUS-E) to detect Gastrointestinal Stromal Tumors (GISTs).

Author	Study	N. GISTs	Lesion Size mm	Echo Pattern	SR/Elastic Scores	Sensitivity	Specificity	Conclusion
Tsuji Y et al., 2016 [62]	Prospective	9 (SELs, *n* = 25)	<20 (36%)20–50 (56%) >50 (8%)	Homogeneous hypoechoic: 2/9 (22.2%); Heterogeneous: 7/9 (77.8%)	Giovannini elastic score 4: 6/9 pts (66.7%); score 5: 3/9 pts (33.3%)	NA	Low	EUS-E may be useful for differentiating GISTs from other SELs; GISTs are characterized as “hard” tissues in comparison to other SELs
Ignee A et al., 2017 [50]	Prospective	57 (SELs, *n* = 62)	62.6 ± 42.1 (16–200)	Blue pattern: 61/62 (98%; Homogenous: 48/61 (79%); Heterogeneous: 13/61 (21%)	No quantification techniques were employed (SR or histogram analysis)	Low	Low	EUS-E is ineffective for distinguishing GISTs from GI leiomyoma because both types of GI mesenchymal tumors are relatively hard lesions
Antonini F et al., 2018 [73]	Retrospective	30 patients	NA	NA	NA	81.8%	85.7%	EUS-E, with a cut-off of 11.18, showed promise in distinguishing GISTs from leiomyomas
Kim SH et al., 2020 [63]	Prospective	7 (SELs, *n* = 31)	23 ± 7	Homogeneous hypoechoic: 7/7 (100%)	SR: 51.1 (29.0–67.0)	100%	94.1%	EUS-E could be a useful diagnostic tool for evaluating gastric SELs, especially in differentiating GISTs from leiomyomas
Guo J et al., 2021 [66]	Retrospective	47	NA	NA	4 channels’ mean hue values of RGB, R, G, and B: 20.25 ± 0.72, −0.79 ± 0.78, 20.79 ± 1.68, and 39.72 ±1.30	50%	78.7%	There was insufficient evidence to support the use of quantitative EUS-E for the differential diagnosis of GIST and leiomyomas

Abbreviations: GISTs: Gastrointestinal stromal tumors; SR: Strain ratio; EUS-E: Endoscopic ultrasound elastography; SELs: Subepithelial lesions; GI: gastrointestinal.

**Table 4 cancers-15-01285-t004:** Studies on the use of Artificial Intelligence (AI) in conjunction with Endoscopic Ultrasound (EUS) for the detection of Gastrointestinal Stromal Tumors (GISTs).

Author	Study	N. EUS Images GISTs	N. GISTs	AI System	Lesion Size mm	Sensitivity	Specificity	AUROC	Conclusion
Kim YH et al., 2020 [102]	Retrospective	905 images of gastric mesenchymal tumors (GIST, leiomyoma, and schwannoma): training dataset; 212 images of gastric mesenchymal tumors: valdation	Training dataset: 125 (69.8%); test dataset: 32 (46.4%)	CNN-CAD system	Training dataset: 3.6 ± 2.1; Test dataset: 3.2 ± 1.6	83.0 (77.4–87.5)	75.5 (69.3–80.8)	79.2 (73.3–84.2)	The CNN-CAD system performed exceptionally well with respect to detecting gastric mesenchymal tumors.
Oh CK et al., 2021 [103]	Retrospective	376 images (*n* = 114 pts)	Training dataset: 85; validation dataset: 54	CNN-based object	25 (10–70)	100% (per-patient)	85.7% (per-patient)	96.3% (per-patient)	High diagnostic ability for predicting gastric GISTs and outperformed human assessment.
Hirai K et al., 2022 [104]	Retrospective	16,110 images (*n* = 631 pts)	Training dataset: 287 (68.5); validation dataset: 63 (70.0); test dataset: 85 (69.7)	AI—deep learning	Training: 25 (2.2–180); validation: 28 (6–130); test: 26.1 (3–180)	98.8%	67.6%	89.3%	In terms of diagnostic performance, the AI system that classified SELs outperformed the experts and may help improve SEL diagnosis in clinical practice.
Yang X et al., 2022 [105]	Retrospective	10,439 images (*n* = 752 pts)	36	AI-based system	Endosonographers’ accuracy in diagnosing GISTs or GI leiomyomas increased from 73.8% (95%CI 63.1–82.2%) to 88.8% (95%CI 79.8–94.2%; *p* = 0.01)				An AI-based EUS diagnostic system was developed that can effectively distinguish GISTs from GI leiomyomas and improve the diagnostic accuracy of SEL assessment.
Tanaka H et al., 2022 [106]	Retrospective	10,600 images (*n* = 53 pts)	42	AI—deep learning involving a residual neural network and leave-one-out cross-validation	26.4	The sensitivity, specificity, and accuracy of AI for diagnosing GISTs were 90.5%, 90.9%, and 90.6%, which can be compared to 90.5%, 81.8%, and 88.7%, respectively, obtained for blind reading (*p* = 0.683)			The diagnostic ability of AI-evaluated CH-EUS results to distinguish between GISTs and leiomyomas was comparable to blind reading by expert endosonographers.
Liu XY et al., 2022 [107]	Meta-analysis (8 studies)	NA	339 (training, validation, and test datasets)	Convolutional neural network (CNN) model	In terms of sensitivity (0.93 vs. 0.71), specificity (0.81 vs. 0.69), and AUC (0.94 vs. 0.75), AI-aided EUS outperformed expert-conducted EUS				AI-assisted EUS is a promising and dependable method for separating SELs with excellent diagnostic performance

Abbreviations: GISTs: Gastrointestinal stromal tumors; SELs: Subepithelial lesions; GI: gastrointestinal; CNN-CAD: Convolutional neural network computer-aided diagnosis.

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
