# Peer review of "Endoscopic Ultrasound Advanced Techniques for Diagnosis of Gastrointestinal Stromal Tumours"

_cancers, 2023, doi:10.3390/cancers15041285_

Round 1

Reviewer 1 Report

This is a review article on the differentiation of GISTs from other benign tumors on EUS. The authors include the previous results of EUS-guided tissue acquisition, elastography, CE-EUS, AI analysis, and so on.  

There are several issues to be considered.

(1) The title is "Enhanced Endoscopic Ultrasound for diagnosis of Gastrointestinal Stromal Tumors". According to the title, the main content should be CE-EUS. Therefore,  the title should be revised.

(2) Tables on previous studies of each diagnostic modality are needed.

(3) Representative images showing elastography, AI analysis, and so on are needed.

(4) More content on AI and CE-EUS is thought to be needed.

Author Response

Response to Reviewer 1

Comments and Suggestions for Authors

This is a review article on the differentiation of GISTs from other benign tumors on EUS. The authors include the previous results of EUS-guided tissue acquisition, elastography, CE-EUS, AI analysis, and so on.  

There are several issues to be considered.

1)The title is "Enhanced Endoscopic Ultrasound for diagnosis of Gastrointestinal Stromal Tumors". According to the title, the main content should be CE-EUS. Therefore, the title should be revised.

Response: We thank the referee for evaluating the paper and its helpful comments. We fully agree with this comment. To be consistent with the manuscript, according to the reviewer’s suggestion, we modified the title as follows: “Endoscopic Ultrasound advanced techniques for diagnosis of Gastrointestinal Stromal Tumours”.

2) Tables on previous studies of each diagnostic modality are needed.

Response: We thank the referee for this helpful suggestion. We have now included 3 tables that summarize studies of each diagnostic modality (Table 2,3,4).

3) Representative images showing elastography, AI analysis, and so on are needed.

Response: We have now included a figure (Figure 3) showing representative images of elastography.

4) More content on AI and CE-EUS is thought to be needed.

Response: We thank the referee for this comment. A new paragraph (Paragraph 6) and a new table (Table 4) summarizing data on artificial Intelligence and CE-EUS have now been included.

Reviewer 2 Report

Pallio et al. wrote a comprehensive and well-organized review on the differential diagnosis of GISTs through EUS. However, some points may be addressed to further improve the quality of the text.

1)      First, I suggest improving English style in order to make it easier and quicker to understand. Moreover, there are some grammar mistakes or missing words. I mention for example:

·         Ligne 60: “Despite the prognosis is mainly associated with the tumor size” should be corrected in “Although the prognosis is mainly associated with the tumor size”

·         Lignes 119-120: “conventional endoscopic forceps biopsy diagnostic yield is limited despite the use of special jaws such as “jumbo” or “bite-on-bite” biopsy and it ranges from 17% to 59%”

·         Lignes 124-125: “After incising the target mucosal and submucosal tissues with an endoscopic submucosal dissection knife using electrosurgical current

·         “Tumour” or “tumor”? “Color” or “colour”? % or “percent”? please be consistent in the whole text

2)      Please carefully review the text, because there are many typos, lacking spaces, missing plurals and wrong/missing punctuation. Please specify what abbreviations stand for at the first time they appear in the text and pay attention not to change abbreviations along the text (e.g Contrast-enhanced harmonic EUS: CH-EUS or CE-EUS? Please be consistent)

3)      A table summarizing EUS features which help distinguishing GISTs from benignant lesions may be useful

Author Response

Response to Reviewer 2

Comments and Suggestions for Authors

Pallio et al. wrote a comprehensive and well-organized review on the differential diagnosis of GISTs through EUS. However, some points may be addressed to further improve the quality of the text.

1) First, I suggest improving English style in order to make it easier and quicker to understand. Moreover, there are some grammar mistakes or missing words. I mention for example:

We thank the referee for evaluating the paper and for its helpful comments. The manuscript has been carefully checked and underwent extensive language editing by a native speaker.

- Ligne 60: “Despite the prognosis is mainly associated with the tumor size” should be corrected in “Although the prognosis is mainly associated with the tumor size”

Response: We thank the referee for this helpful suggestion. The sentence has been corrected as suggested (Page 2, Line 67).

- Lignes 119-120: “conventional endoscopic forceps biopsy diagnostic yield is limited despite the use of special jaws such as “jumbo” or “bite-on-bite” biopsy and it ranges from 17% to 59%”

Response: The sentence has been corrected as suggested (Page 3, Lines 128, 129).

- Lignes 124-125: “After incising the target mucosal and submucosal tissues with an endoscopic submucosal dissection knife using electrosurgical current”.

Response: The sentence has been corrected as suggested (Page 3, Lines 131, 133).

- “Tumour” or “tumor”? “Color” or “colour”? % or “percent”? please be consistent in the whole text.

Response: All the mentioned words have been edited to be consistent throughout the manuscript.

2) Please carefully review the text, because there are many typos, lacking spaces, missing plurals and wrong/missing punctuation. Please specify what abbreviations stand for at the first time they appear in the text and pay attention not to change abbreviations along the text (e.g Contrast-enhanced harmonic EUS: CH-EUS or CE-EUS? Please be consistent).

Response: The text has been carefully reviewed, and all typos, lacking spaces, missing plurals, wrong/missing punctuation, and abbreviations have been checked and corrected.

3) A table summarizing EUS features which help distinguishing GISTs from benignant lesions may be useful.

Response: We thank the referee for this helpful suggestion. We have now included a table that summarizes EUS features which help distinguish GISTs from benignant lesions (Table 1).